# Sondheimer oscillations as a probe of non-ohmic flow in WP$_2$ crystals

Maarten R. van Delft [1✉], Yaxian Wang[2], Carsten Putzke [1], Jacopo Oswald[3], Georgios Varnavides[2], Christina A. C. Garcia [2], Chunyu Guo [1], Heinz Schmid [3], Vicky Süss[4], Horst Borrmann [4], Jonas Diaz[1], Yan Sun [4], Claudia Felser[4], Bernd Gotsmann [3], Prineha Narang [2✉] & Philip J. W. Moll [1✉]

As conductors in electronic applications shrink, microscopic conduction processes lead to strong deviations from Ohm's law. Depending on the length scales of momentum conserving ($l_{MC}$) and relaxing ($l_{MR}$) electron scattering, and the device size ($d$), current flows may shift from ohmic to ballistic to hydrodynamic regimes. So far, an in situ methodology to obtain these parameters within a micro/nanodevice is critically lacking. In this context, we exploit Sondheimer oscillations, semi-classical magnetoresistance oscillations due to helical electronic motion, as a method to obtain $l_{MR}$ even when $l_{MR} \gg d$. We extract $l_{MR}$ from the Sondheimer amplitude in WP$_2$, at temperatures up to $T \sim 40$ K, a range most relevant for hydrodynamic transport phenomena. Our data on $\mu$m-sized devices are in excellent agreement with experimental reports of the bulk $l_{MR}$ and confirm that WP$_2$ can be microfabricated without degradation. These results conclusively establish Sondheimer oscillations as a quantitative probe of $l_{MR}$ in micro-devices.

[1] Laboratory of Quantum Materials (QMAT), Institute of Materials (IMX), École Polytechnique Fédérale de Lausanne (EPFL), Lausanne, Switzerland.
[2] Harvard John A. Paulson School of Engineering and Applied Sciences, Harvard University, Cambridge, MA, USA. [3] IBM Research Europe - Zurich, Rüschlikon, Switzerland. [4] Max Planck Institute for Chemical Physics of Solids, Dresden, Germany. ✉email: maarten.vandelft@epfl.ch; prineha@seas.harvard.edu; philip.moll@epfl.ch

In macroscopic metallic wires, the flow of electric current is well described by Ohm's law, which assigns a metal a spatially uniform 'bulk' conductivity. The underlying assumption is that the complex and frequent scattering events of charge carriers occur on the microscopic length scale of a mean free path, which is much smaller than the size of the conductor, $d$, leading to diffusive behavior. In addition to the scattering processes of bulk systems, the resistance of microscopic conductors is mostly dominated by boundary scattering, thereby masking the internal scattering processes of the bulk in resistance measurements. Here, we present a method to uncover these bulk processes in micro-scale metals, which are of technological importance for fabrication of quantum electronic devices, and simultaneously critical to a fundamental understanding of microscopic current flow patterns. It is instructive to classify the bulk scattering processes into two categories: those that relax the electron momentum, such as electron–phonon, Umklapp or inelastic scattering, occurring at length-scale $l_{MR}$; and those that conserve the electron momentum, such as direct or phonon-mediated electron–electron scattering, associated with a length-scale $l_{MC}$.

Within a kinetic theory framework, these three length scales, namely $d$, $l_{MR}$, and $l_{MC}$, can be used to describe the current flow in micro-scale conductors. When momentum-conserving interactions are negligible, ohmic flow at the macro-scale ($l_{MC} \gg d \gg l_{MR}$) gives way to ballistic transport in clean metals where $l_{MR}$, $l_{MC} \gg d$. Conversely, when momentum-conserving interactions occur over similar or smaller length scales to momentum-relaxing interactions, a third regime of 'hydrodynamic' transport ($l_{MR} \gg d \gg l_{MC}$) is observable[1,2]. In this regime, the static transport properties of electron fluids can be described by an effective viscosity that captures the momentum diffusion of the system[2,3]. These electron fluids exhibit classical fluid phenomena such as Poiseuille flow, whereby the current flow density is greatly decreased at the conductor boundary. Recently, advances in both experimental probes and theoretical descriptions have enabled direct observation of these effects using spatially resolved current density imaging, and have hinted towards the rich landscape of electron hydrodynamics in micro-scale crystals[3–5].

While such local-probe experiments provide means of quantifying electron–electron interactions, and thus extracting $l_{MC}$, direct measurement of the intrinsic momentum-relaxing processes ($l_{MR}$) within micron-scale conductors remains elusive, yet is greatly needed. From a practical perspective, $l_{MR}$ describes the overall scattering from impurities and the lattice vibrations within the metallic microstructure, which at low temperature is an important feedback parameter of quality control in fabrication. Furthermore, given both the reduction of sample size and the improved crystal quality, seemingly exotic transport scenarios where $l_{MR} \gg d \gg l_{MC}$ is satisfied are expected to become more prevalent in technology. An accurate description of these length scales is necessary to predict the overall resistance and thus voltage drops and heat dissipation in the nanoelectronic devices. For example, the resistive processes in a hydrodynamic conductor occur at the boundaries rather than homogeneously distributed in the bulk, which alters the spatial distribution of Joule heating and thereby has significant impacts on thermal design.

Real devices will operate at some intermediate state in the $d$, $l_{MR}$, and $l_{MC}$ parameter space, departing from the well-understood limiting cases of ohmic, ballistic, and hydrodynamic flow. Rich landscapes of distinct hydrodynamic transport regimes are predicted depending on the relative sizes of the relevant length scales[6]. Effective understanding, modeling and prediction of transport requires an experimental method to estimate these parameters reliably in every regime. In large, ohmic conductors, the bulk mean free path $l_{MR}$ can be simply estimated from the device resistance using a Drude model. Yet when $l_{MC}$, $l_{MR} \gtrsim d$,

boundary scattering dominates the resistance, and hence estimates of the bulk scattering parameters are highly unreliable. This leaves the worrying possibility of misinterpreting the transport situation in a conductor, in that the microfabrication itself may introduce defects or changes in the bulk properties that remain undetected by macroscopic observables such as the resistance, but have profound impact on the microscopic current distribution. These effects are already noticeable in state-of-the-art transistors, owing to the low carrier density of semi-conductors[7], but have similarly been reported in metallic conductors[5]. With the increased technological interest in quantum and classical electronics operating at cryogenic temperatures, such questions about unconventional transport regimes are also of practical relevance in next generation electronics[8].

In this context, we propose to exploit a magneto-oscillatory phenomenon, Sondheimer oscillations (SO), as a self-consistent method to obtain the transport scattering length $l_{MR}$ in-situ, even in constricted channels when $l_{MR} \gg d$. In general, a magnetic field (**B**) applied perpendicular to a thin metal forces the carriers on the Fermi surface (FS) to undergo cyclotron motion. Those on extremal orbits of the FS are localized in space due to the absence of a net velocity component parallel to the magnetic field. These localized trajectories can become quantum-coherent, and their interference causes the well-known Shubnikov-de Haas (SdH) oscillations. The states away from extremal orbits also undergo cyclotron motion, yet they move with a net velocity along the magnetic field, analogous to the helical trajectories of free electrons in a magnetic field (Fig. 1). These states are responsible for the Sondheimer size effect which manifests itself as a periodic-in-$B$ oscillation of the transport coefficients, as discovered in the middle of the past century for clean elemental metals[9].

For any given state, the magnitude of **B** sets the helical radius and thus determines how many revolutions the electron completes while traveling from one surface to the other in a micro-device. If an integer number of revolutions occurs, the charge carrier will have performed no net motion along the channel, and hence is semi-classically localized (Fig. 1a). However, if the number of revolutions is non-integer, a net motion along or perpendicular to the channel exists, delocalizing the carriers, resulting in oscillatory longitudinal and transverse magneto-transport behavior. Large-angle bulk scattering events dephase the trajectory, hence the strong sensitivity of SO to the bulk $l_{MR}$ even in nanostructures. These SO are an inherent property of mesoscale confined conductors in three dimensions and have no counterpart in 2D metals like graphene.

The period of the SO is derived by considering a classical charged particle on a helical trajectory between two surfaces perpendicular to the magnetic field[10]. One compares the time it takes to travel the distance $d$ between the surfaces, $t_d = d/v_\parallel$, to the time to complete a single cyclotron revolution, $\tau_c = 2\pi/\omega_c = 2\pi m^*/eB$ ($m^*$: effective mass, $e$: electron charge, $\omega_c = eB/m^*$: cyclotron frequency). Their ratio describes the number of revolutions of the trajectory. For certain fields the helix is commensurate with the finite structure and the number of revolutions is integer, $n$, such that $t_d = n\tau_c$. This occurs periodically in field, with the period given by:

$$\Delta B = \frac{2\pi m^* v_\parallel}{ed} = \frac{\hbar}{ed}\left(\frac{\partial A}{\partial k_\parallel}\right). \qquad (1)$$

The useful identity $v_\parallel = \frac{\hbar}{2\pi m^*}\left(\frac{\partial A}{\partial k_\parallel}\right)$, derived by Harrison[11], directly relates the SO period to the FS geometry, where $v_\parallel$ and $k_\parallel$ denote the velocity and momentum component parallel to the magnetic field and $A$ is the FS cross-sectional area encircled by

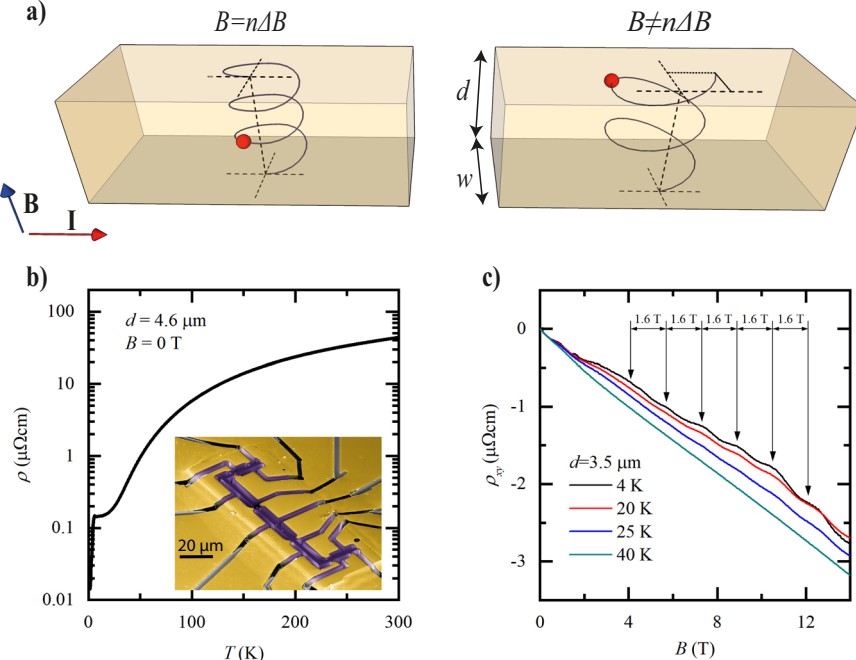

**Fig. 1 Introduction to Sondheimer oscillations. a** Illustration of the Sondheimer effect. Left: the applied magnetic field is $B = 3\Delta B$ and the electron (red) makes an integer number of rotations, with no contribution to transport. Right: $B \neq n\Delta B$. The electron hits the top surface at a different position than its origin on the bottom surface, leading to a contribution to the conductivity. **b** Resistivity as function of temperature for a WP$_2$ microdevice. Inset: false-color SEM image of a typical device used in this study. **c** Sondheimer oscillations seen in the Hall resistivity of a WP$_2$ microdevice, for different temperatures. The oscillation period of $\Delta B = 1.6$ T is highlighted.

the orbit in k-space. Note the contrast to conventional QO which appear around extremal orbits, where $\frac{\partial A}{\partial k_\parallel} = 0$.

All conduction electrons undergo cyclotron motion, yet depending on $\frac{\partial A}{\partial k_\parallel}$, they experience different commensurability fields with a structure of given size $d$. Hence oscillatory contributions to the total conductivity are washed out, unless a macroscopic number of states share the same $v_\parallel \propto \left(\frac{\partial A}{\partial k_\parallel}\right)_{E_f}$[10]. In earlier days of Fermiology[12], geometric approximations for FSs, such as elliptical endpoints, were introduced to identify those generalized geometric features that lead to extended regions of constant $\frac{\partial A}{\partial k_\parallel}$. The computational methods available nowadays allow a more modern approach to the problem. FSs calculated by ab-initio methods can be numerically sliced in order to calculate their cross-section $A(k_\parallel)$. We propose to extend this routine procedure, used to find extremal orbits relevant for QO $\left(\frac{\partial A}{\partial k_\parallel} = 0\right)$, to identify SO-active regions $\left(\frac{\partial^2 A}{\partial k_\parallel^2} \sim 0\right)$, based on the Fermi-surface slicing code SKEAF[13] (see Methods for details on implementation).

SO are caused by the real-space motion of charge carriers and hence also pose some conditions on the shape of the conductor. First, surface scattering needs to be mostly diffusive. If an electron undergoes specular scattering $N$ times before scattering diffusively, it contributes towards the SO as if the sample had an effective thickness $Nd$[14], leading to overtones. Naturally, SO vanish in the (unrealistic) limit of perfectly specular boundary conditions, as such ideal kinetic mirrors remove any interaction of the electron system with the finite size of the conductor. Secondly, the conductor must feature two parallel, plane surfaces perpendicular to the magnetic field to select only one spiral trajectory over the entire structure. The parallelicity requirement is simply given by a fraction of the pitch of the spiral at a certain field (maximal thickness variation $\Delta d < v_\parallel \tau_c = d\frac{\Delta B}{B}$)[10]. These requirements are naturally fulfilled in planar electronic devices.

It is instructive to briefly compare SO to the more widely known QO of resistance, the SdH effect. Both are probes of the FS geometry based on cyclotron orbits, yet the microscopics are strikingly different. While QO frequencies are exclusively determined by FS properties via the Onsager relation and are thus independent of the sample shape, SO are finite-size effects. SO emerge from extended regions on the FS, unlike SdH oscillations to which only states in close vicinity of extremal orbits contribute. While SdH oscillations are quantum interference phenomena, SO are semi-classical, which is key to their use as a robust probe of exotic transport regimes. If both can be observed, powerful statements on the scattering microscopics can be made, as SdH is sensitive to all dephasing collision events and SO separates out the large-angle ones[15]. However, the much more stringent conditions of phase coherence in SdH severely limit their observations at higher temperatures. SO are observable up to relatively high temperatures at which the rapidly shrinking $l_{MR}(T)$ leads to a transition into an ohmic state, when $l_{MR}(T) < d$. As such, they are ideally suited to explore the exotic transport regimes in which, for example, hydrodynamic effects occur.

We apply these theoretical considerations to experimentally investigate the scattering mechanisms in micron-sized crystalline bars of the type-II Weyl semimetal WP$_2$[16] exploiting the Sondheimer effect. Bulk single crystals of WP$_2$ are known for their long $l_{MR}$, in the range of 100–500 μm[17–19], comparable to the elemental metals in which SO were initially discovered[20–23]. These are an ideal test case for non-ohmic electron flow, as hydrodynamic transport signatures and nontrivial electron–phonon dynamics have been observed in various topological semimetals[17,18,24–26]. These ulta-pure crystals are then reduced in size by nanofabrication techniques into constricted channels, to study hydrodynamic or ballistic corrections to the current flow.

Here we employ Focused Ion Beam (FIB) micromachining[27], which allows precise control over the channel geometry in 3D. In

this technique, we accelerate Xe ions at 30 kV to locally sputter the target crystal grown by chemical vapor transport (CVT)[19,28] until a slab of desired dimensions in the μm-range remains. This technique leads to an amorphized surface of around 10 nm thickness, yet has been shown to leave bulk crystal structures pristine[29]. Naturally, reducing the size of a conductor even without altering its bulk mean free path significantly changes the device resistance at low temperatures due to finite size corrections[30]. Hence, measurements of the constricted device resistance alone cannot exclude the possibility of bulk degradation due to the fabrication. Thus far, one could only argue based on size-dependent resistance studies that the values smoothly extrapolate to the bulk resistivity value in the limit of infinite device size[18,31]. Measuring SO directly in the microfabricated devices themselves, however, quantitatively supports that the ultra-high purity of the parent crystal remains unchanged by our fabrication. We note that the fundamental question of the bulk parameters is universal in mesoscopic conducting structures irrespective of the fabrication technique, and these considerations are thus equally applicable to structures obtained by mechanically or chemically thinned samples as well as epitaxially grown crystalline films. SO should provide general insights into the material quality in the strongly confined regime, allowing to contrast different fabrication techniques.

## Results

We measure our μm-confined devices using standard lock-in techniques with applied currents between 50 and 100 μA, low enough to limit self-heating, and magnetic fields up to 18 T. At high temperatures, the measured resistivity agrees well with previous reports on high quality bulk crystals, as expected given the momentum-relaxing limited mean free path of charge carriers in this regime (Fig. 1b). Yet in the low temperature limit, the device resistance exceeds that of bulk crystals by more than an order of magnitude[17,19,32]. Conversely, the residual resistance ratios in our devices (RRR ≈ 160–300) are also considerably lower than in bulk crystals[32]. The main question we address by SO is whether this excess resistance points to fabrication-induced damage, finite size corrections, or a mixture thereof. At low temperatures around 3 K, a drop in resistance signals a superconducting transition. As WP$_2$ in bulk form is not superconducting, this likely arises from an amorphous W-rich surface layer due to the FIB fabrication similar to observations made in NbAs[33] and TaP[34]. In Fig. 1c, we show the Hall resistivity, $\rho_{xy}$, of one of our devices as a function of the magnetic field, for different temperatures. The Hall signal comprises oscillations with a period of $\Delta B = 1.6$ T, resolved above approximately $B = 2$ T.

### The staircase device

A hallmark signature of SO is their linear frequency dependence on the device thickness perpendicular to the field. For this reason, we fabricated crystalline devices with multiple sections of different thickness to study the dependence on the channel thickness, $d$, in a consistent manner. This 'staircase' device allows the simultaneous measurement of transport on 5 steps, as illustrated in Fig. 2. SO appear in all transport coefficients, magnetoresistance (MR) and Hall effect alike, yet here we focus on the Hall effect for two practical reasons. First, the step edges induce non-uniform current flows, and hence the device would need to be considerably longer to avoid spurious voltage contributions from currents flowing perpendicular to the device in a longitudinal resistance measurement. Second, WP$_2$ exhibits a very large MR yet a small Hall coefficient, as typical for compensated semi-metals. Therefore, the SO are more clearly distinguishable against the background in a Hall measurement, but they are also present in the longitudinal channel.

The fabrication process of our WP$_2$ devices follows largely the same procedure as described in ref. [27]. However, for the staircase device, a few key changes were made. In the first fabrication step, the FIB is used to cut a lamella from a bulk WP$_2$ crystal. One side is polished flat, and the other side polished into five sections, each to a different thickness (Fig. 2b). It is then transferred, flat side down, into a drop of araldite epoxy on a sapphire substrate and electrically contacted by Au sputtering (Fig. 2b). In a second FIB step, the staircase slab is patterned laterally into its final structure (Fig. 2a). We use Xe ions for the entire FIB fabrication process in order to avoid potential issues with Ga ion implantation leading to changes in the carrier density. Indeed, experimentally, we see no indication of any charge carrier modulation.

All segments of the staircase devices show pronounced $B$-periodic oscillations in the Hall channel, from which the linear background is removed by taking second derivatives. (Fig. 3). At the lowest fields, a weak, aperiodic structure is observed. In this regime, the cyclotron diameter does not fit into the bar, preventing the formation of the Sondheimer spirals. Note that in all devices of different thickness, this onset field of the SO is the same. This is a natural consequence of the fact that the lateral size, perpendicular to the magnetic field, by design, is the same for all steps of the staircase. Each step, however, differs in thickness $d$ parallel to the magnetic field, and the period varies accordingly between steps (Fig. 3b). At even higher fields, the onset of regular SdH oscillations hallmarks a transition into a different quantized regime. The SO frequency $F = 1/\Delta B$ varies linearly with $d$ as expected (Fig. 3c, Eq. (1)).

### Sondheimer oscillations

Next we identify the Sondheimer-active region on the FS from the ab-initio band structure, which was calculated by density functional theory (DFT) with the projected augmented wave method as implemented in the code of the Vienna ab-initio Simulation Package[35]. The FS of WP$_2$ consists of two types of spin-split pockets: dogbone-shaped electron pockets and extended cylindrical hole pockets (see Fig. 4 and Supplementary Fig. 2 for a complete picture of the FS).

Only one area quantitatively agrees with the observed SO periodicity: the four equivalent endpoints of the dogbone (colored orange in Fig. 3f). Slicing all Fermi-surfaces using SKEAF[13], their cross-sections $A(k_\parallel)$ are obtained. While in QO analysis this information is discarded once the extremal orbits are identified, it forms the basis of the SO analysis. As the dogbone is sliced from the endpoints, the area continuously grows until the two endpoint orbits merge and the area abruptly doubles. Slicing further, the area grows until the maximum orbit along the diagonal is reached. The mirror symmetry of the FS enforces then a symmetric spectrum when slicing further beyond the maximum. The quasi-linear growth at the endpoints signals an extended area of Sondheimer-active orbits. Averaging the near-constant derivative in this region, $\frac{\partial A}{\partial k_\parallel}$, provides via Eq. (1) a tuning-parameter-free prediction of the thickness dependence of the SO frequency. This ab-initio prediction (red line in Fig. 3c) is in excellent agreement with the observed thickness dependence.

Next the temperature-dependence of the SO amplitude is used to gain direct information about the microscopic scattering processes acting on this region of the FS. In Fig. 4a, b, we plot this temperature dependence and highlight two regimes: that of quantum coherence and that of purely SO. In the first regime, quantum coherence leads to SdH oscillations; however, for typical effective masses $m^* \approx m_e$, as in WP$_2$, they are only observable at very low temperatures ($T < 5$ K). Importantly, their quick demise upon increasing temperature is not driven by the temperature dependence of the scattering time, but rather by the broadening of the Fermi-Dirac distribution. This is apparent as their temperature

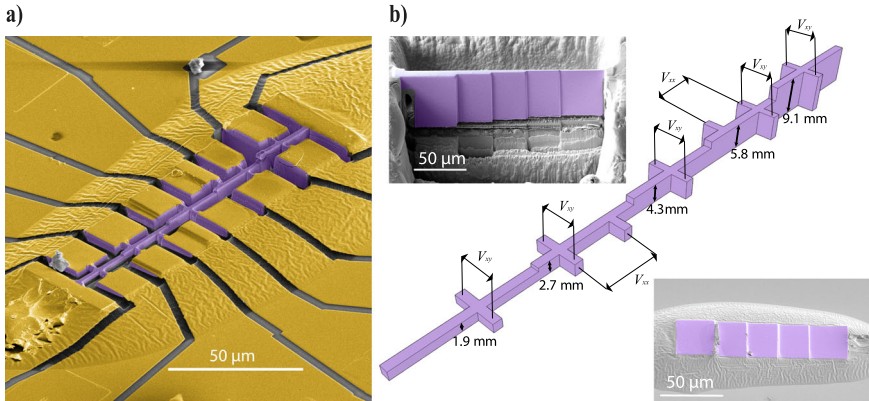

**Fig. 2 The staircase device. a** False color SEM image of a staircase device, used to measure Sondheimer oscillations for different thicknesses. The crystal is colored in purple, and gold contacts in yellow. **b** Main: Schematic of the staircase device, illustrating all possible measurement configurations as well as the thickness of each section. Top left: SEM image of the lamella that will become the device shown in **a**, prior to extracting it from its parent crystal. Bottom right: SEM image of the same lamella, glued down onto a sapphire substrate, ready to define the device geometry. The lamella and glue are covered in gold (not colored) throughout the full field of view. The magnetic field is applied perpendicular to the structure, aligned along the crystallographic [011] direction.

dependence is well described by the Lifshitz–Kosevich formalism based on a temperature-independent quantum lifetime, $\tau_q$.

This strong temperature-suppression of QO severely limits their use to probe scattering mechanisms at elevated temperatures. SO, on the other hand, do not rely on quantum coherence and are readily observed to much higher temperatures, up to 40 K in WP$_2$, while their temperature decay allows a direct determination of the transport lifetime, $\tau_{MR} = l_{MR}/v_F$. Hence SO make an excellent tool to study materials in the temperature range pertinent to exotic transport regimes like ballistic or hydrodynamic. They self-evidence non-diffusive transport as they only vanish when $l_{MR} \sim d$, and hence are only absent in situations of conventional transport within a given device.

## Discussion

Key to observable SO is that electrons do not undergo large-angle scattering events on their path between the surfaces. We therefore have the condition that $l_{MR} > d$[36,37]. As $l_{MR}(T)$ decreases with increasing temperature and the boundary scattering is assumed to be temperature-independent, the SO amplitude is suppressed as $e^{-d/l_{MR}(T)}$ which allows us to estimate the bulk transport mean free path within a finite-size sample, even when $d \ll l_{MR}$. It is extracted as[36]:

$$\frac{1}{l_{MR}(T)} = -\frac{1}{d} \ln \frac{A(T)}{A(0)}, \tag{2}$$

where $A(T)$ is the SO amplitude at temperature $T$. $A(T = 0)$ is estimated by extrapolation, which is a robust procedure as the SO amplitude saturates at low but finite temperatures. This is analogous to the saturation of the resistivity of bulk metals at low temperatures, once bosonic scattering channels are frozen out and temperature-independent elastic defect scattering becomes dominant.

In the following discussion, we focus on the scattering time $\tau_{MR}$ to facilitate comparison of our results with literature and theory, using the average Fermi velocity on the dogbone FS determined from our band structure calculations self-consistently, $v_F = 3.6 \times 10^5$ m/s. The $\tau_{MR}(T)$ obtained from all devices quantitatively agrees, despite their strong difference in thickness (between 1.3 and 4.6 μm) and hence SO frequency, further supporting the validity of this simple analysis (see Fig. 4c and Supplementary Fig. 5). The lifetimes on the SO devices furthermore agree with measurements on bulk crystals[18], evidencing that the increased

resistivity compared to bulk can be wholly attributed to finite size corrections rather than to any fabrication-induced damage, and that FIB fabrication does not introduce significant changes to the bulk properties of WP$_2$ that might cause misinterpretations of the scattering regime.

For our WP$_2$ devices, a standard Dingle analysis[15] of the QO yields a quantum scattering time $\tau_q \sim 10^{-13} - 10^{-12}$ s (Fig. 4c), in agreement with published values for bulk crystals WP$_2$[19]. As $\tau_q$ is sensitive to all dephasing scattering events, but $\tau_{MR}$ only to large-angle momentum-relaxing scattering, the microscopics of the scattering processes in WP$_2$ are brought to light. The four orders of magnitude difference between $\tau_{MR}$ and $\tau_q$ reflects a common observation in topological semi-metals such as Cd$_3$As$_2$[38], PtBi$_2$[39], or TaAs[40].

Long $\tau_{MR}$, together with a high quality, clean sample, enables the realization of the hydrodynamic regime where the momentum-conserving scattering dominates. These quantitative measurements of $\tau_q$ and $\tau_{MR}(T)$ can now be directly compared to theoretical models of scattering. We consider an initial electronic state with energy $\varepsilon_{n\mathbf{k}}$ (where $n$ and $\mathbf{k}$ are the band index and wavevector respectively) scattering against a phonon with energy $\omega_{\mathbf{q}\nu}$ (where $\nu$ and $\mathbf{q}$ are the phonon polarization and wavevector respectively), into a final electronic state with energy $\varepsilon_{m\mathbf{k}+\mathbf{q}}$. The electron–phonon scattering time $\tau_{e-ph}$ describing such an interaction can be obtained from the electron self energy using Fermi's golden rule:

$$
\begin{aligned}
\tau_{e-ph}^{-1}(n\mathbf{k}) = &\frac{2\pi}{\hbar} \sum_{m\nu} \int_{BZ} \frac{d\mathbf{q}}{\Omega_{BZ}} |g_{mn,\nu}(\mathbf{k}, \mathbf{q})|^2 \\
&\times \left( n_{\mathbf{q}\nu} + \frac{1}{2} \mp \frac{1}{2} \right) \delta \left( \varepsilon_{n\mathbf{k}} \mp \omega_{\mathbf{q}\nu} - \varepsilon_{m\mathbf{k}+\mathbf{q}} \right),
\end{aligned}
\tag{3}
$$

where $\Omega_{BZ}$ is the Brillouin zone volume, $f_{n\mathbf{k}}$ and $n_{\mathbf{q}\nu}$ are the Fermi–Dirac and Bose–Einstein distribution functions, respectively, and the electron–phonon matrix element for a scattering vertex is given by

$$g_{mn,\nu}(\mathbf{k}, \mathbf{q}) = \left( \frac{\hbar}{2m_0 \omega_{\mathbf{q}\nu}} \right)^{1/2} \langle \psi_{m\mathbf{k}+\mathbf{q}} | \partial_{\mathbf{q}\nu} V | \psi_{n\mathbf{k}} \rangle. \tag{4}$$

Here $\langle \psi_{m\mathbf{k}+\mathbf{q}} |$ and $| \psi_{n\mathbf{k}} \rangle$ are Bloch eigenstates and $\partial_{\mathbf{q}\nu} V$ is the perturbation of the self-consistent potential with respect to ion displacement associated with a phonon branch with frequency

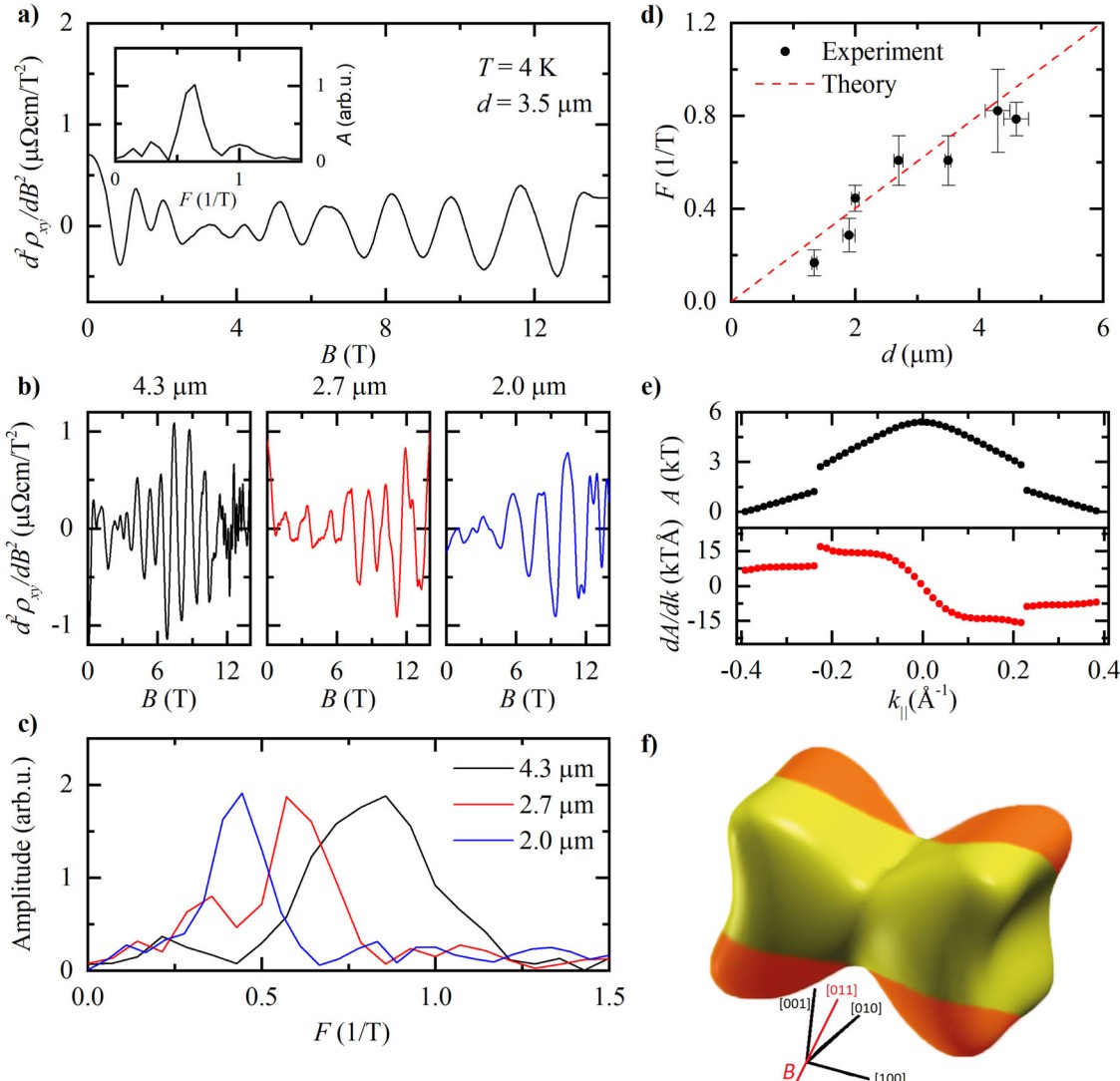

**Fig. 3 Analysis of Sondheimer oscillations in WP$_2$. a** Second derivative of the Hall resistivity shown in Fig. 1c at $T = 4$ K. Inset: Fast Fourier Spectrum (FFT) corresponding to this data. **b** Second derivatives of the Hall resistivity at three different thicknesses, $d = 4.3$, 2.7 and 2.0 μm, $T = 4$ K. **c** FFTs corresponding to the data in **b**. **d** Dependence of the Sondheimer frequency on $d$. The error bars in the frequency, $F$, are derived from the width of the relevant peaks in the FFT spectra and those in $d$ from the standard deviation of thickness measurements made with SEM. The red dashed line is calculated from the Fermi surface as determined from DFT. **e** Cross-sectional area, $A$, of the dogbone Fermi surface pocket of WP$_2$ as a function of $k$ parallel to the field direction of our experiments (top), and its derivative (bottom). **f** Location of observed Sondheimer orbits drawn in orange on the dogbone-shaped Fermi surface pocket. The magnetic field is applied along the [011]-direction, perpendicular to the current, as indicated by the red line.

$\omega_{\mathbf{q}\nu}$. Plotting these state-resolved electron–phonon lifetimes at ~10 K on the FS reveals the distribution of scattering in the SO-active regions (Fig. 4d). Equation (3), however, accounts, to first order, for all electron–phonon interactions, irrespective of the momentum transfer or equivalently the scattering angle. To remedy this, we augment the scattering rate with an 'efficiency' factor[41] given by the relative change of the initial and final state momentum $(1 - \frac{v_{n\mathbf{k}} \cdot v_{n\mathbf{k}}}{|v_{n\mathbf{k}}||v_{n\mathbf{k}}|} = 1 - \cos\theta)$, where $v_{n\mathbf{k}}$ is the group velocity and $\theta$ is the scattering angle:

$$\left(\tau_{\text{e--ph}}^{\text{mr}}(n\mathbf{k})\right)^{-1} = \frac{2\pi}{\hbar} \sum_{m\nu} \int_{\text{BZ}} \frac{d\mathbf{q}}{\Omega_{\text{BZ}}} \left|g_{mn,\nu}(\mathbf{k}, \mathbf{q})\right|^2$$
$$\times \left(n_{\mathbf{q}\nu} + \frac{1}{2} \mp \frac{1}{2}\right) \delta\left(\varepsilon_{n\mathbf{k}} \mp \omega_{\mathbf{q}\nu} - \varepsilon_{m\mathbf{k}+\mathbf{q}}\right) \quad (5)$$
$$\times \left(1 - \frac{v_{n\mathbf{k}} \cdot v_{n\mathbf{k}}}{|v_{n\mathbf{k}}||v_{n\mathbf{k}}|}\right).$$

At low temperatures, the thermally activated phonon modes have a tiny **q**, therefore the initial and final electronic states only differ from a small angle. It is thus important to take this momentum-relaxation efficiency factor into account in addition to $\tau_{\text{e--ph}}$, in order to estimate $\tau_{\text{MR}}$ which determines the electron mean free path in the SO-active regions. In the SO measurements, the electron orbits are located on the endpoints of the dogbone-shaped electron pockets (Fig. 3f), therefore we highlight the scattering efficiency distribution on the electron FS in Fig. 4e. Indeed, when the orbit is aligned along the diagonal direction, the FS cross section features very low scattering efficiency with an averaged $1 - \cos\theta < 0.1$. This supports our observation of frequently scattering electrons with long transport lifetimes in the SO measurement.

These results demonstrate the power of the Sondheimer size effect for the extraction of the momentum-relaxing mean free path in mesoscopic devices when $d \ll l_{\text{MR}}$ via their temperature

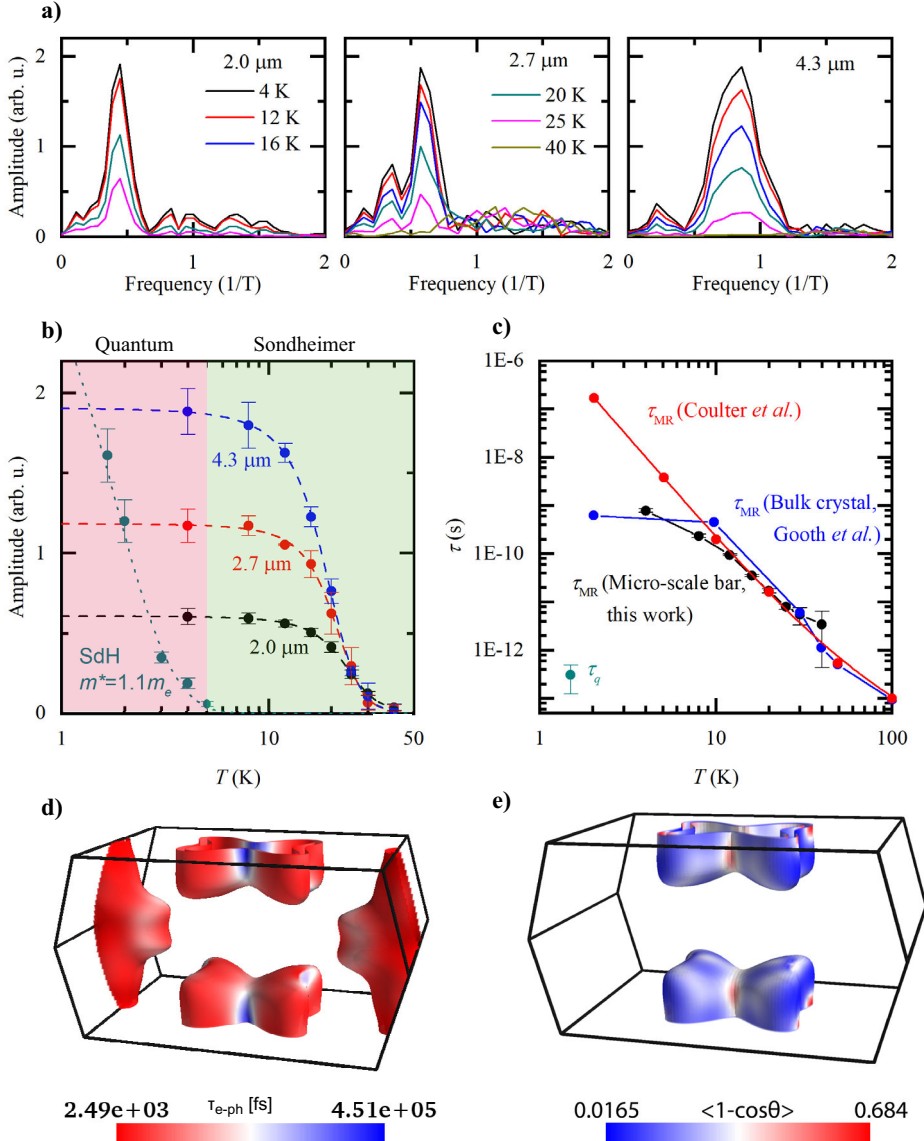

**Fig. 4 Extraction of scattering times from the Sondheimer amplitude. a** FFTs of the SO at different temperatures for thicknesses of 4.3, 2.7, and 2.0 μm. The data at $T = 4$ K is the same as that in Fig. 3c. **b** Temperature dependence of the Sondheimer and SdH oscillation amplitudes, for different sample thicknesses. The error bars are estimated from the variation in amplitude in the FFT spectra. The dashed lines are fits used to extrapolate to the amplitude at zero temperature, $A(0)$ (see Methods for details). The dotted line is a Lifshitz–Kosevich fit, giving an effective mass of $1.1\,m_e$. Two regimes are highlighted: that of quantum coherence, where SdH oscillations exist alongside SO, and that of Sondheimer, where only SO exist. **c** Scattering times extracted for a 4.3 μm thick section of a WP$_2$ device using Eq. (2) and the calculated Fermi velocity, $v_F = 3.6 \times 10^5$ m/s. An approximate quantum lifetime extracted from the SdH oscillations as well as data from refs. [18,46] are included for comparison. Errors in $\tau_{MR}$ are propagated from those in the amplitudes and the error in $\tau_q$ is the standard deviation of several measurement. **d** Calculated scattering time for all electron–phonon scattering ($\tau_{e-ph}$) and **e** the scattering efficiency determining the momentum-relaxing scattering lifetimes ($\tau_{MR}$) projected onto the Fermi surface at $T = 10$ K.

dependence. Combined with first-principles theoretical calculations, we were able to locate the states contributing to the helical motion to the elliptical endpoints of a particular FS of WP$_2$. We note, however, that such analysis as well as the thickness dependence are only relevant for the academic purpose of robustly identifying these oscillations as SO. Once this is established, the relevant lifetimes may straightforwardly be obtained from the resistance oscillations at a single thickness. Hence, SO promise to be a powerful probe to obtain the bulk mean free path in devices with μm-scale dimensions without relying on any microscopic model assumptions. This analysis is a clear pathway to identify scattering processes in clean conductors within operating devices. It thereby provides important feedback of the materials quality

after a micro-/nano-fabrication procedure and disentangles the roles of bulk and surface scattering that are inseparably intertwined in averaged transport quantities of strongly confined conductors, such as the resistance. As their origin is entirely semi-classical, they are not restricted by stringent criteria such as quantum coherence and thus span materials parameters of increased scattering rate. In particular, they survive up to significantly higher temperatures and thereby allow microscopic spectroscopy in new regimes of matter dominated by strong quasiparticle interactions, such as hydrodynamic electron transport. With this quantitative probe, it will be exciting to test recent proposals of exotic transport regimes and create devices that leverage such unconventional transport in quantum materials.

## Methods

**Crystal growth.** High quality crystals of WP₂ were grown via the CVT method using iodine as transport agent, with the following starting materials: red phosphorus (Alfa-Aesar, 99.999%) and tungsten trioxide (Alfa-Aesar, 99.998%). The starting materials were sealed in an evacuated fused silica ampoule. A two-zone furnace with a temperature gradient of 1000–900 °C was used for the CVD method. After several weeks, the ampoule was removed from the furnace and quenched in water. The crystals were characterized by X-ray diffraction.

**Fabrication of the staircase device.** In Fig. 2, we have shown the so-called staircase device which was especially designed for the measurement of SO. Here, we describe its fabrication in detail, beginning from a bulk crystal. Firstly, the crystallographic orientation of the bulk crystal is determined through XRD measurements, and it is glued in the desired orientation onto an SEM stub. It is then introduced into a Helios G4 PFIB UXe DualBeam FIB/SEM using Xe ions, hence avoiding issues associated with surface implantation as in the more common Ga-based sources. The focused beam of Xe ions is used to cut a rectangular slab we call a lamella from that crystal in three steps. The first is using a high 2.5 μA current to cut two gaps into the crystal, separated by an ~15 μm thick and 150 μm long section of crystal. A smaller current of 0.2 μA is then used to smoothen this crystal section in order to produce moderately flat sidewalls, after which the wall is cut near the bottom and on the sides such that we have a lamella attached to the parent crystal through two beams at its top.

At this stage, a 15 nA current is used to fine-cut both sides of the lamella, leaving it with parallel sidewalls and a high level of smoothness. The thickness of the lamella is now that of what will be thickest section of the staircase device. The lamella is then divided into five sections; the three middle sections are of equal length, while the sections on either end are longer for further contacts. Starting from one side, we leave the first section untouched and polish and mill the other sections to their respective target thickness using cleaning-cross-sections as the milling strategy. This is the stage shown in the top left insert of Fig. 2b. After this, we cut through the beam on one side and thin down the other side.

An ex-situ micromanipulator is used to break the thinned beam and pick up the stepped lamella. On a sapphire substrate with lithographically prepatterned gold contact pads, we place a small droplet of Araldite epoxy. While the epoxy is still liquid, we place our lamella, flat side down, onto the droplet. Capillary forces then create a profile of epoxy around the lamella that extends smoothly up to each of its top surfaces, without covering any. After curing the epoxy for 1 h at 150 °C, we take the substrate to a sputtering machine, where it is briefly RF etched and 3 nm of Ti plus 200 nm of Au are sputtered onto the lamella, glue and prepatterned gold contacts through a shadow mask. The lamella after this step is shown in the bottom right of Fig. 2b.

In order to pattern the device, we again make use of the Helios G4 PFIB. The Ti/Au layer that covers what will be the active part of the device is first removed with an acceleration voltage of 5 kV and an ion current of 2 nA. The overall shape of the device with the position of the contacts is then cut out at 0.3 nA and 30 kV and the central bar of the device is gently polished in order to create smooth sidewalls. Finally, the Ti/Au layer away from the device is cut through in order to separate the contact pads and make sure that current can only flow through the device, which is then ready for measurement.

In order to check that the crystallographic orientation of the final device is as expected based on the initial XRD measurements, we perform measurements of the MR. The MR has a characteristic shape with a minimum for **B**∥**c** and a maximum for **B**∥**b**[19], allowing an identification of these axes. In Supplementary Fig. 1, we show an angle-dependent measurement of MR for one of our devices, which is indeed aligned 45° away from the *b* and *c* axes.

**Calculation of $A(k)$ and $\frac{dA}{dk}$.** The calculations of the *k*-dependent cross-sectional area of the FS shown in Fig. 3 and Supplementary Fig. 3 were performed with a slightly modified version of SKEAF[13], a commonly used software designed to numerically extract QO frequencies from calculated band structures. QO take place at extremal areas of the FS (i.e. when $\frac{dA}{dk} = 0$) and their frequency relates to the extremal area as given by the Onsager relation[15]: $F = \frac{\hbar}{2\pi e} A$. This correspondence allows us to display $A(k)$ in units of kT rather than $1/\text{Å}^2$. As QO are more known than SO, this choice of unit facilitates their comparison.

The SKEAF algorithm, written in the Fortran 90 language, reads electronic structures calculated by DFT in the Band-XCrySDen-Structure-File format. It constructs a cubic super cell much larger than the original reciprocal unit cell and aligned with the magnetic field direction. This super cell is then divided into slices perpendicular to the magnetic field, and the software records the cross-sectional area for each slice. During regular use, SKEAF then matches the orbits over the different slices and finds the extremal ones. We, however, need the area for each slice, and hence we have added a short section of code to create a new file containing the *k*-values and areas (in both $1/\text{Å}^2$ and kT) for all orbits.

This file contains many copies of each FS sheet. Rather than averaging each orbit, as is done by SKEAF for the extremal orbits, we simply select one copy and plot this as in Fig. 3e. This is reasonable, as the differences between the areas of the copies are consistently less than 0.1%. Finally, we take a numerical derivative of $A(k)$ and find $\frac{dA}{dk}$, from which we can identify possible Sondheimer orbits.

**Extraction of $l_{MR}$ from the SO amplitude.** We follow a step-by-step procedure in order to extract the $l_{MR}$ as a function of temperature. In the first step, the measured longitudinal or Hall resistivity is smoothed and differentiated twice. The required level of smoothing is adjusted for each dataset to the extent that no oscillatory component of the data is removed, while the noise is suppressed. Importantly, the same procedure is performed consistently for each temperature. After taking the second derivative, we perform an FFT of the data using a Hanning window. The relevant amplitude $A(T)$ is found from the peak in the FFT and plotted against the temperature (see Fig. 4b). In order to extract $A(0)$, we then need to extrapolate to $T = 0$ K. In order to do this, we use a fit of the form $A(T) = A_1 / \left(1 + \exp\left(\frac{T - c1}{c2}\right)\right)$, which provides an excellent empirical description of the data and allows us to determine $A(0)$. As the SO amplitude saturates at low temperatures, the exact extrapolation procedure has little effect on the value of $A(0)$ and the extrapolation is robust.

Finally, we use Eq. (2) to calculate $l_{MR}(T)$, which we plot in Fig. 4b. In Supplementary Fig. 5, we show $l_{MR}(T)$ for several different devices, showing consistency between the values extracted for any thickness, from $\rho_{xx}$ or $\rho_{xy}$ and from measurements along different crystallographic axes.

**Ab initio calculations.** The ab initio calculations were performed with the open source DFT code JDFTx[42]. We used fully relativistic Perdew–Burke–Ernzerhof pseudopotentials[43–45] and included the spin-orbit coupling effect in all calculations. A kinetic energy cutoff of 28 Ha was used along with a $6 \times 6 \times 8\Gamma$-centered *k*-mesh and a Fermi-Dirac smearing of 0.01 Ha for the Brillouin zone integration. Both the lattice constants and the ion positions were relaxed until the energy differences were less than $10^{-9}$ Ha. To compute the electron–phonon scattering time, we performed frozen phonon calculations in a $3 \times 3 \times 2$ supercell, and obtained 44 maximally localized Wannier functions by projecting the plane-wave bandstructure to W *d* and P *p* orbitals, which allowed us to converge the electron scattering calculation on a much finer $66 \times 66 \times 88\mathbf{k}'$ and **q** grid for $T = 10$ K.

## Data availability

The data generated in this study have been deposited in the Zenodo repository, https://doi.org/10.5281/zenodo.4675599. Data presented in Fig. 4d, e are available upon request.

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

## Acknowledgements

M.R.v.D. acknowledges funding from the Rubicon research program with project number 019.191EN.010, which is financed by the Dutch Research Council (NWO). This project was funded by the European Research Council (ERC) under the European Union's Horizon 2020 research and innovation program (grant no. 715730, MiTopMat). Y.W. is partially supported by the STC Center for Integrated Quantum Materials, NSF Grant No. DMR-1231319 for development of computational methods for topological materials. This research used resources of the National Energy Research Scientific Computing Center, a DOE Office of Science User Facility supported by the Office of Science of the U.S. Department of Energy under Contract No. DE-AC02-05CH11231 as well as resources at the Research Computing Group at Harvard University. P.N. is a Moore Inventor Fellow and gratefully acknowledges support through Grant No. GBMF8048 from the Gordon and Betty Moore Foundation. C.A.C.G. acknowledges support from the NSF Graduate Research Fellowship Program under Grant No. DGE-1745303. We acknowledge financial support from DFG through SFB 1143 (project-id 258499086) and the Würzburg-Dresden Cluster 274 of Excellence on Complexity and Topology in Quantum Matter - ct.qmat (EXC 2147, project-id 39085490). B.G. acknowledges financial support from the Swiss National Science Foundation (grant numner CRSII5_189924). H.S and B.G thank J. Gooth for discussion and K. Moselund, S. Reidt, and A. Molinari for support, and received funding from the European Union's Horizon 2020 research and innovation program under Grant Agreement ID 829044 "SCHINES".

## Author contributions

M.R.v.D., C.P., J.O., C.G., J.D. performed the transport experiments, as well as the microfabrication in collaboration with B.G. and H.S. The crystals were grown by V.S. and C.F., and crystallographically analyzed by H.B. Y.S. and C.F. calculated the band structures, and Y.W., G.V., C.A.C.G., P.N. performed the electron–phonon scattering calculations. B.G., C.F., P.N., and P.J.W.M. conceived the experiment, and all authors contributed to writing of the manuscript.

## Competing interests

The authors declare no competing interests.
