## [Peer Review File · Nature Communications]

REVIEWER COMMENTS

Reviewer #1 (Remarks to the Author):

The present manuscript reports exploitation of Sondheimer oscillations as a method to obtain I_{MR} in micro/nano-devices. Basically, this report firmly establishes the methodology of Sondheimer oscillations, which is utilized in the analysis of electrical conduction of a micro/nano-device and allows for identification of its conduction regime and the extraction of conduction length scales. This methodology also overcomes the limitations of the analysis based on the Drude model in the micro/nano-devices and of SdH oscillations. Thus, the present study is of significance and widespread usage of this methodology as a tool to characterize the conduction mechanisms is expected to be made in the field of micro/nano-devices. As the manuscript is novel and no error was found in the analysis, I recommend the publication of the present manuscript in Nature Communications. A few typos should be corrected before publication.

Reviewer #2 (Remarks to the Author):

The paper by van Delft et al exploits Sondheimer oscillations (SO) to obtain the momentum relaxing mean-free path in microdevices. This is important for evaluating the quality of the quantum electronic devices after microfabrication; a step that can be harmful for intrinsic material properties. The evaluation of mean-free-path can also help to assign a transport regime (ohmic, ballistic or hydrodynamic), which the device operates in. The authors take WP2 material and fabricate a staircase device using focused ion beam technique. Such device geometry helps evaluating SO and the mean free path, which is in agreement with the bulk values of WP2. This serves as evidence that WP2 can be processed with FIB without degradation and the device can operate in a hydrodynamic regime. The paper can be interesting from the technical point of view, since it demonstrates evidences that WP2 can have good quality after FIB fabrication. However, the evaluation of the mean-free-path from SO is already well documented and the authors cite the previous works properly. Therefore, the proposal of this manuscript to use SO for mean-free-path evaluation is clearly not novel. The hydrodynamic transport regime of the fabricated device is also not demonstrated.

New components of this work is the observation of SO oscillations in WP2 and, what I find appealing, is Fig 4c comparing different scattering times. Whether this is sufficient for publication in Nat Communications or is more suitable for a specialized community, I would like to leave it up to the Editor's decision.

1) I find the title is misleading, since it suggests that the type-II Weyl semimetal property is important for the effects reported here. However, the topological characteristics of the material are not a part of SO treatment; only a long I_{MR} is important. Whether the Weyl physics survives in FIB fabricated devices is yet to be demonstrated. The title "Sondheimer oscillations in FIB fabricated WP2 crystal" can better match the content of the presented manuscript.

2)The emergence of oscillating R_{xy} behavior is not explained. The concept Figure 1a is confusing and picks up only a particular electron velocity that moves on a spiral along the magnetic field. It looks like the device is insulating at this particular magnetic field, since the electrons are localized.

3)It is unclear what frequency in Fig. 4a is taken for the analysis of scattering time. The notations for y-axis in Figures 4a and 4b are different, though these should be the same quantities.

4)It would be interesting to add in Fig. 4c the scattering time evaluated from the sample resistance at

zero field.

5) I missed the discussion of the spectrum in Fig. 4a. What do the different frequencies correspond to?

6) How large is the electrical current in the measurements? Do the devices have ohmic characteristics? The device dimensions reaching the regime when the size quantization can start become important. Did the author consider this possibility?

7) The device dimensions should be given consistent throughout the paper. Sometimes the device dimension is given as 4.6 μm , in another figure it is 4.2 μm or even 4.3 μm . This inconsistency appears for other sizes as well. Are these different devices? And what devices are used for evaluating the mean free path?

8) Does the FIB fabrication change the charge carrier density? Is the charge carrier density the same in all segments of the staircase device?

Reviewer #3 (Remarks to the Author):

In their manuscript 'Sondheimer oscillations as a probe of non-ohmic flow in type-II Weyl semimetal WP2'. The authors report magneto-resistance measurements on high-quality micromachined WP2 crystals at various temperatures, aiming to discern different electronic transport regimes, namely ballistic and hydrodynamic.

Using Sondheimer oscillation, they extract the electron momentum-relaxing scattering times, which: (1) in the ballistic regime is not limited by the device width and (2) is applicable at temperatures where hydrodynamic electron flow is claimed to appear. The measured electron scattering rates at high temperatures agree with a phonon-mediated hydrodynamic transport theory and previous studies (Gooth, J. et al.). However, no new information on the microscopic origin of hydrodynamic behavior in WP2 is provided.

Nevertheless, the methodology presented in the manuscript for obtaining quantitative information on the momentum relaxing scattering times in working devices is essential when probing non-ohmic electron flow, particularly for any claim of hydrodynamic behavior - which is of current interest in the field.

The paper is easy to read and clearly written. The conclusion that Sondheimer oscillations are useful in situ probe for non-ohmic electron transport in 3D conductors is convincing and will interest the community.

Therefore, I recommend publication subject to a few comments/clarifications:

(1) It would be instructive for the reader to compare the I_{mr} extracted from resistivity and Hall measurements to the SO extracted I_{mr} to clearly show the SO method advantage.

(2) The maximum temperature probed appears to be 40 K for both figure 4 and S5. However, in the abstract and text, it is claimed 50 K. What is the maximum temperature you observe Sondheimer oscillations?

(3) In Fig. 4C, the extracted scattering time is for which device thickness?

(4) In Fig. S5, there is a missing legend for one of the data sets.

(5) Line 29 in supplementary. The units of current are mislabeled 0.2 μm , should be 0.2 μA .

Reviewer #1 (Remarks to the Author):

The present manuscript reports exploitation of Sondheimer oscillations as a method to obtain I_{MR} in micro/nano-devices. Basically, this report firmly establishes the methodology of Sondheimer oscillations, which is utilized in the analysis of electrical conduction of a micro/nano-device and allows for identification of its conduction regime and the extraction of conduction length scales. This methodology also overcomes the limitations of the analysis based on the Drude model in the micro/nano-devices and of SdH oscillations. Thus, the present study is of significance and widespread usage of this methodology as a tool to characterize the conduction mechanisms is expected to be made in the field of micro/nano-devices. As the manuscript is novel and no error was found in the analysis, I recommend the publication of the present manuscript in Nature Communications. A few typos should be corrected before publication.

Thank you for your encouraging words on our work!

Reviewer #2 (Remarks to the Author):

The paper by van Delft et al exploits Sondheimer oscillations (SO) to obtain the momentum relaxing mean-free path in microdevices. This is important for evaluating the quality of the quantum electronic devices after microfabrication; a step that can be harmful for intrinsic material properties. The evaluation of mean-free-path can also help to assign a transport regime (ohmic, ballistic or hydrodynamic), which the device operates in. The authors take WP2 material and fabricate a staircase device using focused ion beam technique. Such device geometry helps evaluating SO and the mean free path, which is in agreement with the bulk values of WP2. This serves as evidence that WP2 can be processed with FIB without degradation and the device can operate in a hydrodynamic regime. The paper can be interesting from the technical point of view, since it demonstrates evidences that WP2 can have good quality after FIB fabrication. However, the evaluation of the mean-free-path from SO is already well documented and the authors cite the previous works properly. Therefore, the proposal of this manuscript to use SO for mean-free-path evaluation is clearly not novel. The hydrodynamic transport regime of the fabricated device is also not demonstrated.

New components of this work is the observation of SO oscillations in WP2 and, what I find appealing, is Fig 4c comparing different scattering times. Whether this is sufficient for publication in Nat Communications or is more suitable for a specialized community, I would like to leave it up to the Editor's decision.

Sondheimer oscillations pose a classical phenomenon in clean metals in high magnetic fields and have been discovered long ago. We have taken care, as noted, to cite the relevant works and give credits to the scientists discovering them. Please allow us to discuss the novel aspects which go well beyond the classical literature of SO. SO by itself is a rather exotic niche phenomenon that

is not well-known outside of the expert community, unlike quantum oscillations (dHvA, SdH) which are taught in undergraduate courses. The simple reason is that while QO proved pivotal in developing our understanding of metals and their Fermi surfaces; SO failed to provide a use case. It had been proposed as a spectroscopic technique to learn about critical endpoints of the Fermi surface, yet it failed to deliver accuracy and to show new insights to be learned about a metal from the detailed knowledge of these curvature terms. The crux is of course the sensitivity of SO on the sample shape, the surface termination, and the roughness, which bulk quantization of QO is immune to. Attempts shifted to use it as a surface spectroscopy technique, determining the specular coefficient, and even use it as sensors for adsorbed gasses on the surface. While SO by themselves were interesting manifestations of semi-classical dynamics of electrons at the time, none of this, to our knowledge, contributed significantly to our understanding of metals.

This is the main conceptual point we make here. With the current developments of non-ohmic current flows, such as hydrodynamics and its playforms, the bulk mean free path and its relation to dynamic scattering scales (el-ph; el-el;...) as well as to the finite size of the metal becomes a critical parameter. Here, we point to the so-far overlooked critical advantage of SO as they sense the bulk mean-free-path even in strongly confined microdevices. This important transport length scale is commonly very different from the quantum mean free path probed by QO, largely due to the immunity of the former to small-angle scattering events. In classical SO experiments on bulk metallic crystals, the extracted scattering time is found to be highly consistent with the scattering times obtained from transport. This does not work anymore in complex and confining shapes. Pointing to this, and demonstrating an actual quantitative and model-parameter-free determination of a scattering mean free path more than 2 orders of magnitude larger than the size of the device, are the main observations going beyond the established SO literature.

1) I find the title is misleading, since it suggests that the type-II Weyl semimetal property is important for the effects reported here. However, the topological characteristics of the material are not a part of SO treatment; only a long l_{MR} is important. Whether the Weyl physics survives in FIB fabricated devices is yet to be demonstrated. The title "Sondheimer oscillations in FIB fabricated WP2 crystal" can better match the content of the presented manuscript.

This is of course a matter of taste. Please let us lay out the reasoning for our choice of title and why we prefer to keep the original title. The title you propose is a correct technical summary of the experiment, it states what has been done. Yet it does not convey the implications of the work, the meaning of the results, or their relevance for current research fields. Our target audience are researchers interested in novel transport regimes beyond ballistic/diffusive in solids. The manuscript is explicit in that the Weyl-II properties are not directly relevant to the observation. Yet mentioning it we feel is extremely helpful to point readers interested in this materials class to this work, making them aware of the experimental possibilities as well as inspiring theoretical works that could address SO in Weyl systems. Non-ohmic flow is a key research area of transport in topological systems, while the question of whether or not topology is a factor in obtaining hydrodynamic transport is open.

Whether the Weyl physics survives in FIB fabricated devices is yet to be demonstrated.

The SO are not impacted by Weyl physics, and whether or not to mention it in the title is clearly a matter of taste. However, regarding the science of Weyl-II in these structures, there is no evidence supporting or even suggesting a topological change in the microstructures. We conclusively demonstrate that neither the -very long- bulk mean free path nor the quantum oscillation spectrum is changed by the microfabrication. The micron-size of the structures is well beyond any finite size confinement or quantization effects (see discussion below). Given these observations, there seems little doubt about the survival of the bulk band structure and properties.

2)The emergence of oscillating R_{xy} behavior is not explained.

Although the initial prediction of Sondheimer oscillations did not extend to R_{xy} , it has since been well established experimentally and theoretically that oscillations do occur in R_{xy} as well (see e.g. Munarin *et al.*, Physical Review 172, 718-736 (1968)). As the physical mechanism is the same as for R_{xx} , this is not explained separately in our manuscript. We have now changed our introduction of the origin of oscillatory behavior to mention transverse transport as well.

The concept Figure 1a is confusing and picks up only a particular electron velocity that moves on a spiral along the magnetic field. It looks like the device is insulating at this particular magnetic field, since the electrons are localized.

Indeed, we share that point fully and this figure was already before submission subject of intense discussions by the authors. However, we were not able to come up with a better version. The figure is intended to visualize why the electronic system may show commensurability oscillations. The only way to improve the figure we see involves drawing multiple spirals of different velocities that all match the criterion. This quickly gets crowded, and is also deceptive: Regardless of the Fermi surface shape, any metal features (not necessarily circular) spirals along the field. Yet SO only appear if there is an extended region of constant dA/dk which inevitably is lost in a set of differently sized spirals. This is the reason why we eventually kept this figure as it visualizes commensurability of an electron trajectory, despite all its shortcomings. Given that already in the classical papers on SO such spiral pictures were used (e.g. H.J. Trodahl, J.Phys. C: Solid St. Phys. 13, 1764-1777 (1971)), it may be fair to say that this problem does not have a simple visual solution. We would highly appreciate any concrete ideas how to improve clarity here.

3)It is unclear what frequency in Fig. 4a is taken for the analysis of scattering time. The notations for y-axis in Figures 4a and 4b are different, though these should be the same quantities.

The scattering time plotted in figure 4c was measured in a 4.3 μm thick section of a staircase device (data shown in figures 3b and 4a, left). We have adapted the figure caption to clarify this. As we also demonstrate in figure S5, this scattering time is consistent across several devices, with different thickness and orientation.

Indeed, the y-axes of figures 4a and 4b represent the same quantity, and should have been labelled in the same way. We have corrected this in the manuscript.

4) It would be interesting to add in Fig. 4c the scattering time evaluated from the sample resistance at zero field.

We would love to do this, yet how could this be done? The main issue we address here is that the bulk scattering time is inaccessible by transport measurements. This material is so clean that the sample is 2 orders of magnitude smaller than the mean free path, and its resistance is (almost) entirely due to boundary scattering on a very complex shape – the role of bulk scattering in establishing voltage gradients in transport is negligibly small. The 3D ballistic regime is highly non-local and very difficult to tackle quantitatively, especially given the complex Fermi surface and the complex device shape. Even if that could be done by a Boltzmann approach, there is no hope to identify the few and very rare bulk scattering events in this situation – this is exactly why the SO are so powerful. Certainly, a model calculation is possible at very high temperatures (300 K) when the bulk mean free path is nanoscopic and a diffuse transport situation is established. There, a simple local diffusion equation can be trivially evaluated in any shape numerically. Yet once in the diffuse regime, we do not have any Sondheimer or QO values to compare it to.

5) I missed the discussion of the spectrum in Fig. 4a. What do the different frequencies correspond to?

We are sorry if the presentation caused any confusion. The frequency spectrum of Fig. 4a is exactly the same as in 3c, in which only the $T=4$ K curves of different thickness are plotted. From the position of the peak, the thickness dependence in 3d and the association with the Fermi surface lobes is done. Fig. 4a just shows the temperature dependence of the same data, which allows us to extract the temperature-dependent mean free path discussed in 4b,c. As there is only an amplitude reduction but no change in the spectra, we did not discuss this again. We have added a line into the caption.

6) How large is the electrical current in the measurements? Do the devices have ohmic characteristics? The device dimensions reaching the regime when the size quantization can start become important. Did the author consider this possibility?

Thank you for pointing this out, we have added a statement into the paper. We have used currents of 50 or 100 μA in our measurements. At these currents, the devices respond linearly (no higher harmonic generation), which excludes intrinsic or extrinsic non-linear transport, for example through self-heating. This should have been stated clearly, sorry.

For size quantization, we are still quite large. WP_2 is a metal with Fermi-surfaces spanning a decent fraction of the Brillouin zone. The associated wavelength hence is a small number of unit cells, far from the micron-size we investigate here. Quantization is indeed most interesting, and as we now can confirm the bulk material quality, it is the goal of a separate research project to obtain $<100\text{nm}$ class devices which may enter a quantized regime. This is not simple for various

technical reasons. For example, the heat dissipation under ion irradiation is through the bulk of the target crystal. As one cuts the devices that thin, given that by design they are not coupled to a substrate directly, the thermal conductivity drops rapidly which leads to heat buildup and eventually chemistry. In the devices presented here, the observation of quantum oscillations in good agreement with bulk values excludes a significant band structure modification through quantization.

7)The device dimensions should be given consistent throughout the paper. Sometimes the device dimension is given as 4.6 μm , in another figure it is 4.2 μm or even 4.3 μm . This inconsistency appears for other sizes as well. Are these different devices? And what devices are used for evaluating the mean free path?

The thickness of 4.3 μm corresponds to the middle section of the staircase device shown in figure 2. This was wrongly labelled within this figure as 4.2 μm (we have now corrected this). The thickness of 4.6 μm corresponds to a different device. Altogether, we have measured Sondheimer oscillations for thicknesses of 1.3, 1.9, 2.0, 2.7, 3.5, 4.3 and 4.6 μm . We have evaluated the mean free path for each of these, and found the results to be highly consistent. A statement on the range of thicknesses was added to the main text.

8)Does the FIB fabrication change the charge carrier density? Is the charge carrier density the same in all segments of the staircase device?

We do not see any evidence for charge carrier density changes. We know that there is no degradation of the bulk quantum or transport mean free path, hence it is reasonable not to expect chemical composition / doping changes in the bulk. Furthermore, WP_2 is a high carrier density material, and even if some amount of surface doping was active, it is unrealistic for it to change the bulk carrier density. As a general precaution, we fabricate these devices using a Xe beam. Xe has no implantation potential and completely leaves the sample after irradiation. This addresses a common issue of Ga surface implantation in FIB methods. Yet again, given the high carrier density even surface Ga implantation would be unlikely a source of carrier density variations. Indeed, experimentally we observe the same SdH frequencies of WP_2 bulk in the different segments, speaking against notable carrier density modulation in the devices.

Reviewer #3 (Remarks to the Author):

In their manuscript 'Sondheimer oscillations as a probe of non-ohmic flow in type-II Weyl semimetal WP_2 '. The authors report magneto-resistance measurements on high-quality micromachined WP_2 crystals at various temperatures, aiming to discern different electronic transport regimes, namely ballistic and hydrodynamic. Using Sondheimer oscillation, they extract the electron momentum-relaxing scattering times, which: (1) in the ballistic regime is not limited by the device width and (2) is applicable at temperatures where hydrodynamic electron flow is claimed to appear. The measured electron

scattering rates at high temperatures agree with a phonon-mediated hydrodynamic transport theory and previous studies (Gooth, J. et al.). However, no new information on the microscopic origin of hydrodynamic behavior in WP2 is provided.

Nevertheless, the methodology presented in the manuscript for obtaining quantitative information on the momentum relaxing scattering times in working devices is essential when probing non-ohmic electron flow, particularly for any claim of hydrodynamic behavior - which is of current interest in the field.

The paper is easy to read and clearly written. The conclusion that Sondheimer oscillations are useful in situ probe for non-ohmic electron transport in 3D conductors is convincing and will interest the community.

We highly appreciate your supportive comments to our work, thank you.

Therefore, I recommend publication subject to a few comments/clarifications:
(1) It would be instructive for the reader to compare the I_{mr} extracted from resistivity and Hall measurements to the SO extracted I_{mr} to clearly show the SO method advantage.

We would love to do this, yet how could this be done? The main issue we address here is that the bulk scattering time is inaccessible by transport measurements. This material is so clean that the sample is 2 orders of magnitude smaller than the mean free path, and its resistance is (almost) entirely due to boundary scattering on a very complex shape – the role of bulk scattering in establishing voltage gradients in transport is negligibly small. The 3D ballistic regime is highly non-local and very difficult to tackle quantitatively, especially given the complex Fermi surface and the complex device shape. Even if that could be done by a Boltzmann approach, there is no hope to identify the few and very rare bulk scattering events in this situation – this is exactly why the SO are so powerful. Certainly, a model calculation is possible at very high temperatures (300 K) when the bulk mean free path is nanoscopic and a diffuse transport situation is established. There, a simple local diffusion equation can be trivially evaluated in any shape numerically. Yet once in the diffuse regime, we do not have any Sondheimer or QO values to compare it to.

(2) The maximum temperature probed appears to be 40 K for both figure 4 and S5. However, in the abstract and text, it is claimed 50 K. What is the maximum temperature you observe Sondheimer oscillations?

The maximum temperature at which we observed Sondheimer oscillations was in fact 40 K. The 50 K mentioned in the text was an error that we have corrected in the new version of our manuscript.

(3) In Fig. 4C, the extracted scattering time is for which device thickness? The scattering time plotted in figure 4c was measured in a 4.3 μm thick section of a staircase device (data shown in figures 3b and 4a, left). We have adapted the figure caption to clarify this.

(4) In Fig. S5, there is a missing legend for one of the data sets.

We have added the missing legend.

(5) Line 29 in supplementary. The units of current are mislabeled 0.2 μm , should be 0.2 μA .

We have corrected this mistake. Thank you!

List of changes:

- Caption of Fig. 4c: "Scattering times extracted for WP_2 using Eq. 2 and the Fermi velocity from Ref.18." \rightarrow "Scattering times extracted for a 4.3 μm thick section of a WP_2 device using Eq. 2 and the calculated Fermi velocity, $v_f=3.6*10^5$ m/s."
- Y-axis label of Fig 4b: "A (arb.u.)" \rightarrow "Amplitude (arb.u.)"
- Label in Fig 2b: "4.2 μm " \rightarrow "4.3 μm ".
- Added caption for the brown diamond symbols in figure S5.
- Corrected line 29 of the SI: "0.2 μm " \rightarrow "0.2 μA ".
- In the abstract: "up to $T\sim 50$ K" \rightarrow "up to $T\sim 40$ K".
- Line 188: "up to 50 K" \rightarrow "up to 40 K".
- Lines 70-72: "However, if the number of revolutions is non-integer, a net motion along the channel exists, delocalizing the carriers, resulting in oscillatory magnetotransport behavior." \rightarrow "However, if the number of revolutions is non-integer, a net motion along or perpendicular to the channel exists, delocalizing the carriers, resulting in oscillatory longitudinal and transverse magnetotransport behavior."
- Line 132: "At high temperatures, the resistivity measured in μm -confined devices agrees well..." \rightarrow "We measure our μm -confined devices using standard lock-in techniques with applied currents of 50 or 100 μA , low enough to limit self-heating, and magnetic fields up to 18 T. At high temperatures, the measured resistivity agrees well..."

- Lines 156-158, added: “We use Xe ions for the entire FIB fabrication process in order to avoid potential issues with Ga ion implantation leading to changes in the carrier density. Indeed, experimentally, we see no indication of any charge carrier modulation.”
- Caption Fig. 4a, added: “The data at T=4 K is the same as that in Fig. 3c.”
- Fig. 4a: exchanged labels “2.0 μm ” and “4.3 μm ”.
- Line 204, added: “(between 1.3 and 4.6 μm)”.
- Line 69: “occur” \rightarrow “occurs”.
- Abstract: “semi-metal” \rightarrow “semimetal”.
- Line 134: “those” \rightarrow “that”.
- Caption figure S5: “Device thickness, channel (Hall or longitudinal) and current direction (along the crystallographic a-axis, or between the b and c-axes.” \rightarrow “Device thickness, channel (Hall or longitudinal) and current direction (along the crystallographic a-axis, or between the b- and c-axes) do not affect the measured lifetime.”
- Line 169: “Figs. 3f” \rightarrow “Fig. 3f”.
- SI line 17: “phosphorous” \rightarrow “phosphorus”.
- Lines 19, 50, 123, 134, 196, 235 and 241 and: “mean-free-path” \rightarrow “mean free path”.

REVIEWERS' COMMENTS

Reviewer #2

I would like to thank the authors for addressing my comments and questions. Some issues have been clarified. I am particularly thankful to the authors for discussing the novel aspects of their work, since I thought I may have missed some important arguments.

Let me explain below the important points that I cannot agree.

- The authors point to the “so-far overlooked critical advantage of SO as they sense the bulk mean-free-path even in strongly confined microdevices”. I cannot agree that this is something that has been overlooked. Reference 20 is from year 1979 and it is already stated there that: (i) the SO depends on the electron scattering in the volume, (ii) SO spectrum makes it possible to determine the mean free path. This paper and others make clear how the SO signal emerges. In fact, the nature of SO emergence requires some confinement. (The device size is comparable or smaller than the mean free path.) It is therefore unclear why the authors think that the “confinement” was overlooked. Because of the microstructure and the SO sensitivity to bulk scattering the authors can judge that the FIB seems not to affect the bulk part of WP2, at least in terms of the mean free path.

The value of this MS can be that the authors re-introduce Sondheimer oscillations (40-50 years after their discovery), that remained largely unused so far. This MS does not show new aspects of SO phenomena. Neither the MS demonstrates anything on the hydrodynamic transport features and certainly does not probe non-ohmic flow.

- I also cannot agree with the author's approach on selecting the title. The current title misleads the reader and awakes wrong expectations. A reader anticipates type-II Weyl physics to come into play, but at the end the reader remains disappointed that Weyl is not important here. (Yes, the paper states in the text that Weyl is not important.) Any reference to Weyl is therefore inappropriate here. A material with a sufficiently large mean free path will show the same effect. Neither the MS demonstrates how the SO probes the non-ohmic flow. The SO probes only the mean free path in the bulk.
The authors may anticipate implications of their work and discuss it in the text body. Whether this will then be realized at some point or not, is a different question. It will certainly be interesting to see in the future how the SO can probe the non-ohmic flow.
One may also anticipate that, in the regime of hydrodynamic flow the emergence of SO may be altered, since the electron velocity changes gradually from the wall of the constriction to the bulk. Then the electron trajectory may be not a simple spiral as shown in Fig. 1, but may have some more complicated shape, which may change the oscillation frequency (?). Furthermore, a formation of a superconducting layer in the FIB fabricated WP2 microdevice hinders the studies of topological aspects of device performance, since the current distribution between the bulk and the surface will depend on temperature, magnetic field (?). At this moment it is unclear how the contribution of both can be clearly separated. Non-uniformity of the microdevice given by the amorphous surface can be unfavorable for studying the hydrodynamic flow. Hopefully, the future studies clarify how and whether this limitation can affect the transport phenomena.
- I may have poorly formulated my original question about the spectrum in Figure 4a. Let me rephrase it. Every panel in Fig. 4a contains a dominating frequency and some side peaks with a lower intensity. Can the authors comment on the appearance of those side

peaks? What can their origin be? Do they come from the inhomogeneity of the structure, such as density or structure size variation? Or does it have other origin?

In overall, I find little novelty presented in this work, except the observation of Sondheimer oscillations in an FIB fabricated structures. The work did not give any new insights into SO or non-ohmic flow. In my opinion the work demonstrates an interesting technical progress, but in terms of physics the progress is not evident. Unjustified title misbalances the paper presentation and leaves an impression that the authors attempt to ascribe more value to their work, than it presents at the current stage of research.

With this I would like to leave the decision to the editor whether this MS (after changing the title) can be suitable for the readership of Nature Communications.

Reviewer #3 (Remarks to the Author):

The authors addressed my questions. I have no further comments.

I would like to thank the authors for addressing my comments and questions. Some issues have been clarified. I am particularly thankful to the authors for discussing the novel aspects of their work, since I thought I may have missed some important arguments. Let me explain below the important points that I cannot agree.

- The authors point to the “so-far overlooked critical advantage of SO as they sense the bulk mean-free-path even in strongly confined microdevices”. I cannot agree that this is something that has been overlooked. Reference 20 is from year 1979 and it is already stated there that: (i) the SO depends on the electron scattering in the volume, (ii) SO spectrum makes it possible to determine the mean free path. This paper and others make clear how the SO signal emerges. In fact, the nature of SO emergence requires some confinement. (The device size is comparable or smaller than the mean free path.) It is therefore unclear why the authors think that the “confinement” was overlooked. Because of the microstructure and the SO sensitivity to bulk scattering the authors can judge that the FIB seems not to affect the bulk part of WP2, at least in terms of the mean free path.

Possibly our response was not worded as clearly as it should have. Indeed, it was already known in 1979 that SO require confinement and depend on the mean free path, in that sense confinement certainly was not overlooked but is at the heart of these classical explanations.

However, these old works aimed at explaining the SO as an, at that time, novel physical phenomenon. We see this in complete analogy to quantum oscillations, in which the challenge lied in explaining why the magnetization and resistivity becomes oscillatory in high fields. Yet there is a key difference. As QO allowed to tomographically explore the Fermi surfaces of metals, it has become a cornerstone of our exploration of metals. SO depend on the sample geometry, and on non-extremal cross-section of a Fermi surface – facts that rendered the SO oscillations of little practical use.

The value of this MS can be that the authors re-introduce Sondheimer oscillations (40-50 years after their discovery), that remained largely unused so far. This MS does not show new aspects of SO phenomena. Neither the MS demonstrates anything on the hydrodynamic transport features and certainly does not probe non-ohmic flow.

Here we have a fully orthogonal viewpoint. Naturally SO is insensitive to small-angle MC-scattering and does not directly show hydrodynamic behavior. This has been done elsewhere (Nat. Comm. 9:4093 (2018)). Here we address one of the main concerns in the community with that work, about if and how the scattering is changed by microfabrication. If microstructuring had significantly altered the material, these results may not have been based on confinement, but on sample modification. Before our work, there was no hope to disentangle these two parameters. Now by measuring I_{MR} independently in confined samples, we can settle this important question in the field of hydrodynamic transport. Hence we strongly disagree with the statement that this work does not contribute new insights to hydrodynamic transport in solids – quite the contrary.

- I also cannot agree with the author’s approach on selecting the title. The current title misleads the reader and awakes wrong expectations. A reader anticipates type-II Weyl physics to come into play, but at the end the reader remains disappointed that Weyl is not important here. (Yes, the paper states in the text that Weyl is not important.) Any reference to Weyl is therefore inappropriate here. A material with a sufficiently large mean free path will show the same effect. Neither the MS demonstrates how the SO probes the non-ohmic flow. The SO probes only the mean free path in the bulk.

In our previous response, we have outlined our reasoning for the choice of title. Although we disagree on this point, we accept to change the title and remove the mention of Weyl.

The authors may anticipate implications of their work and discuss it in the text body. Whether this will then be realized at some point or not, is a different question. It will certainly be interesting to see in the future how the SO can probe the non-ohmic flow. One may also anticipate that, in the regime of hydrodynamic flow the emergence of SO may be altered, since the electron velocity changes gradually from the wall of the constriction to the bulk. Then the electron trajectory may be not a simple spiral as shown in Fig. 1, but may have some more complicated shape, which may change the oscillation frequency (?). Furthermore, a formation of a superconducting layer in the FIB fabricated WP2 microdevice hinders the studies of topological aspects of device performance, since the current distribution between the bulk and the surface will depend on temperature, magnetic field (?). At this moment it is unclear how the contribution of both can be clearly separated. Non-uniformity of the microdevice given by the amorphous surface can be unfavorable for studying the hydrodynamic flow. Hopefully, the future studies clarify how and whether this limitation can affect the transport phenomena.

We agree that fully incorporating SO trajectories into hydrodynamic flow is the next challenge, and we work on this currently theoretically. It will remain to be seen how to explore these flow patterns. However, the situation is much more complex than sketched by the referee. The drift velocity v measures the net velocity at a point, in the spirit of a current density $j=nev$. It strongly differs from the velocity of the individual particle that enters the Lorentz force equation and hence leads to the spiral. Given that in a Fermi liquid particles are locked to velocities very close to v_F in metals, one would not expect any sizable variation of the spiral itself. Rather, interspiral collisions will form a steady-state in which the net current density close to the boundary is suppressed – in the meaning of cancelling spirals.

- I may have poorly formulated my original question about the spectrum in Figure 4a. Let me rephrase it. Every panel in Fig. 4a contains a dominating frequency and some side peaks with a lower intensity. Can the authors comment on the appearance of those side peaks? What can their origin be? Do they come from the inhomogeneity of the structure, such as density or structure size variation? Or does it have other origin?

Unlike the main peaks, these side peaks are not reproducible. Regarding their origin, we can exclude variations in the structure. The required difference in size to produce a distinct frequency is large enough that we would be able to clearly observe this in SEM. We also know that our devices are made of high-quality single crystals, and inhomogeneity is unlikely. One should also consider the small window size accessible to experiments, from which low amplitude (sub-)harmonics naturally appear even if only one frequency is present.

In overall, I find little novelty presented in this work, except the observation of Sondheimer oscillations in an FIB fabricated structures. The work did not give any new insights into SO or nonohmic flow. In my opinion the work demonstrates an interesting technical progress, but in terms of physics the progress is not evident. Unjustified title misbalances the paper presentation and leaves an impression that the authors attempt to ascribe more value to their work, than it presents at the current stage of research.

With this I would like to leave the decision to the editor whether this MS (after changing the title) can be suitable for the readership of Nature Communications.